# Pathophysiology-Based Management of Acute Heart Failure

**Luigi Falco, Maria Luigia Martucci, Fabio Valente, Marina Verrengia, Giuseppe Pacileo and Daniele Masarone \*** 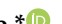

Heart Failure Unit, Department of Cardiology, AORN dei Colli-Monaldi Hospital Naples, 80131 Naples, Italy
\* Correspondence: daniele.masarone@ospedalideicolli.it

**Abstract:** Even though acute heart failure (AHF) is one of the most common admission diagnoses globally, its pathogenesis is poorly understood, and there are few effective treatments available. Despite an heterogenous onset, congestion is the leading contributor to hospitalization, making it a crucial therapeutic target. Complete decongestion, nevertheless, may be hard to achieve, especially in patients with reduced end organ perfusion. In order to promote a personalised pathophysiological-based therapy for patients with AHF, we will address in this review the pathophysiological principles that underlie the clinical symptoms of AHF as well as examine how to assess them in clinical practice, suggesting that gaining a deeper understanding of pathophysiology might result in significant improvements in HF therapy.

**Keywords:** acute heart failure; pathophysiology; tailored treatment

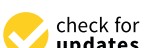



## 1. Introduction

Acute heart failure (AHF) is a clinical syndrome characterized by the rapid or gradual onset of symptoms and/or signs related to heart failure [1]; these symptoms or signs should be significant enough to prompt the patient to seek urgent medical intervention, resulting in unplanned hospitalization or visit to the emergency department [2]. Despite substantial advances in pharmacologic and non-pharmacologic therapy in managing patients with chronic heart failure with marked improvements in long-term survival, rates of rehospitalization at 3 months and mortality at 12 months after an AHF episode remain respectively at 10–30% [3,4]. Although the clinical presentation of AHF is highly variable, the most common reason for hospitalization is significant volume overload and, subsequently, congestive symptoms. Fewer patients present with hypotension and symptoms of reduced organ perfusion [5]. As congestion and hypoperfusion play a central role in the management of AHF and in determining the prognosis, understanding the underlying pathophysiological mechanisms related to them is essential for the appropriate treatment of patients with AHF. Therefore, in this review, we will discuss practically the pathophysiological principles underlying the clinical syndrome of AHF and examine how to evaluate them in clinical practice to promote a tailored pathophysiological-based treatment of patients with AHF.

## 2. Pathophysiology of Congestion

In AHF, there are two main types of congestion [6]:

1.  Peripheral congestion: characterized by a progressive increase in body weight, peripheral edema, jugular distension, hepatomegaly, ascites, and renal venous stasis [7];
2.  Pulmonary congestion: featured by worsening dyspnea, pulmonary rales, and B-lines at lung ultrasound [8].

Peripheral congestion usually coexists with pulmonary congestion, but the reverse is not always true.

These two types of congestion recognize different pathophysiologic mechanisms, whereby peripheral congestion recognizes fluid retention as the primary mechanism (congestion related to cardiac failure) [9]. In contrast, fluid redistribution is the leading cause of pulmonary congestion (congestion related to vascular failure) [10].

### 2.1. Congestion Related to Cardiac Failure

The reduction in cardiac output secondary to myocardial dysfunction results in arterial underfilling that is sensed by mechanoreceptors present in the left ventricle, carotid sinus, aortic arch, and renal afferent arterioles resulting in an increased sympathetic outflow from the central nervous system, activation of the renin-angiotensin-aldosterone system (RAAS) and the nonosmotic release of arginine-vasopressin [11–15].

Activation of these systems, together with increased release of substances with vaso-constrictive activity (e.g., endothelin and vasopressin) and the development of resistance to the action of endogenous natriuretic peptides [16], contribute to the retention of sodium and water that tend to balance (through an increase in cardiac output) adverse effects of AHF on oxygen delivery to the peripheral tissues [17]. However, persistent activation of these systems results in impaired regulation of sodium excretion through the kidneys, which results in sodium and, secondarily, fluid accumulation and tissue edema [7]. Tissue edema develops when the amount of transudate fluid moving from the capillaries to the interstitium exceeds the maximum drainage capacity of the lymphatic system [18]. The transudate of plasma fluid into the interstitium depends on the relationship between on-cotic and hydrostatic pressure in the capillaries and interstitium (Figure 1): increasing the transcapillary gradient of hydrostatic pressure and decreasing the transcapillary gradient of oncotic pressure promotes the formation of interstitial edema [18].

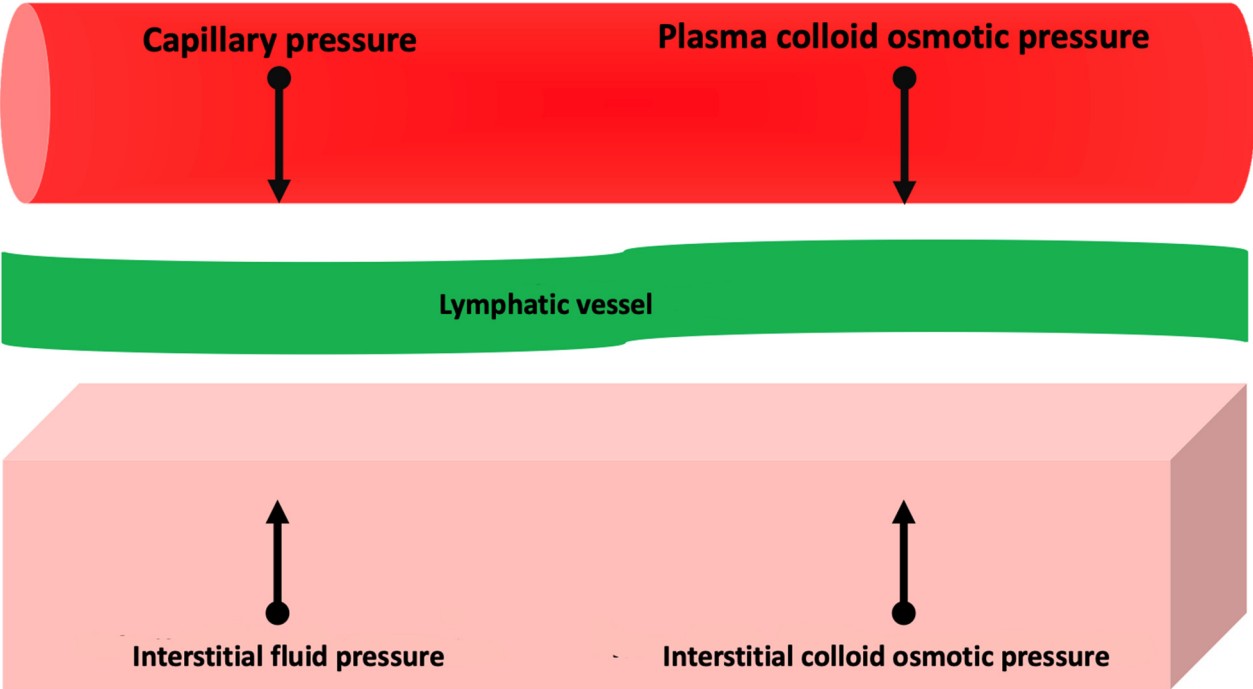

**Figure 1.** Pathophysiology of interstitial edema. See text for further information.

Several studies have elucidated further mechanisms promoting interstitial edema. The impairment of the network of glycosaminoglycans (due to chronic sodium accumulation) of connective tissues, which in the healthy subject can buffer a high amount of reabsorbed sodium, thus preventing compensatory water retention, contributes to edema formation in the patient with AHF [19].

Venous congestion is linked to renal and hepatic dysfunction, which may play a role in edema formation as indicated by different research [20,21]. Historically, worsening renal function in AHF patients was hypothesized as a consequence of a reduced cardiac output resulting in renal hypoperfusion. In contrast, recent data indicate that venous congestion (assessed as increased central venous pressure) is the primary hemodynamic determinant for developing renal dysfunction [22], whereas reduced cardiac output has minor effects

on renal function [23]. Moreover, visceral congestion can increase intra-abdominal pressure in AHF, with further adverse effects on renal function [24,25]. Recent data have indeed shown that reducing central and intra-abdominal venous pressure by decongestive therapy (diuretics, ultrafiltration, paracentesis) can improve glomerular filtrate [26–30].

Regardless of the mechanisms implicated in the onset of acute cardio-renal syndrome, renal dysfunction can exacerbate sodium and fluid retention with, consequently, further, increase in capillary hydrostatic pressure and promotion of interstitial edema formation [31].

Transient hepatic dysfunction is often present in patients with AHF and, in the overwhelming majority of cases, is cholestatic and related to right heart failure [32].

Furthermore, in patients with AHF (particularly in patients with advanced stage of the disease), hepatic dysfunction, together with intestinal congestion, may contribute to a reduction in protein synthesis [33] with a consequent decrease in oncotic capillary pressure that promotes the formation of interstitial edema. Finally, there are plenty of investigations suggesting that venous congestion is not simply an epiphenomenon secondary to cardiac dysfunction but instead plays an active and detrimental role in the pathophysiology of AHF by inducing pro-oxidant [34], pro-inflammatory [35], and hemodynamic stimuli that contribute to the progression of AHF [36]. Previous in vitro studies highlighted endothelium activity in diverse experimental models [37–39]. These observations were further investigated in animal [40] and human models [35,41]. Nitric oxide (NO), prostaglandins (PGs), reactive oxygen species (ROS), and cytokines are just a few of the molecules that endothelium produces. These factors are essential for maintaining a state of stable of chronic heart failure as well as promoting the shift to AHF [42]. Mechanistic insights by which these pathophysiological processes are induced remains poorly understood, but models indicate that biomechanical forces generated in early stages of congestion contribute significantly to endothelial and neurohumoral activation [43,44]. The endothelium works as a master regulator of vascular homeostasis continuously recording its surrounding environment [42]. Indeed, biomechanical stressors as congestion-derived wall stretch and biochemical triggers as increased RAAS activity, are sensed by endothelial cells [45]. Therefore, working as a control system, ECs undergo a phenotypic change toward a pro-oxidant and pro-inflammatory vasoconstriction state [41,46]. These pleiotropic effects have consequences on kidneys, affecting tubuloglomerular feedback [47], and on endothelium itself, increasing the permeability [48]. Thus, the vicious cycle of peripheral congestion is continued [42].

### 2.2. Congestion Related to Vascular Failure

Fluid accumulation alone cannot explain the entire pathophysiology of congestion in AHF; in fact, most patients with AHF have only a slight increase in body weight (<1 kg) before the onset of clinical symptoms [49].

In these patients, congestion is precipitated predominantly by fluid redistribution rather than fluid accumulation [50].

Indeed, it is well known that adrenergic stimulation results in a transient vasoconstriction that leads to a sudden displacement of fluids from the splanchnic and peripheral venous system to the pulmonary circulation in the absence of exogenous fluid retention [51,52]. However, the prerequisite for that mechanism to be realized is the pre-existence of some degree of peripheral and splanchnic congestion (albeit minimal).

Under physiological conditions, the capacitating veins contain about 25% of the circulating volume and, through a dampening of volume overload, induce stabilization of cardiac preload [53]. In hypertensive-based AHF, the mismatch in the ventricular-vascular coupling relationship due to an increase in afterload and an increase in preload by vasoconstriction of the capacitance veins results in the appearance of pulmonary edema [8].

Both fluid accumulation and redistribution are responsible for congestion during an AHF episode, but their significance depends on the patient profile. While fluid accumulation represents the primary mechanism of peripheral congestion in patients with worsening heart failure with reduced ejection fraction [17], fluid redistribution represents the predominant pathophysiologic mechanism in de novo vascular type AHF in patients

with preserved ejection fraction [54]. Consequently, therapy aimed at resolving congestion should be individualized. While in patients with fluid accumulation, diuretics should be the drugs of choice [55], on the other hand vasodilators are the most appropriate drugs for restoring ventricular-vascular coupling in patients with fluid redistribution [8].

A detailed description of the clinical parameters, ultrasonographic data, and biomarkers used to identify congestion in patients with AHF is beyond the scope of this review. However, the main elements used in clinical practice for the identification of pulmonary and peripheral congestion are summarized in Table 1.

**Table 1.** Clinical, echocardiographic, and laboratory parameters used for the assessment of congestion in clinical practice. JVP: Jugular Venous Pulsation. HF: Heart Failure NT-proBNP: N-terminal fragment pro B-type Natriuretic Peptide.

| Parameters | Peripheral Congestion | Pulmonary Congestion | Notes |
|---|---|---|---|
| JVP > 8 cm | Yes | No | Difficult to assess (particularly in obese patients) |
| Hepatomegaly | Yes | No | Also due to non HF causes |
| Bilateral legs edema | Yes | No | Also due to non HF causes |
| Rales with base-apex gradient | No | Yes | Also due to non HF causes |
| Bendopnea | Yes | Yes | Also due to non HF causes |
| Inferior vena cava collapse < 50% with sniff | Yes | No | Difficult to assess in positive pressure ventilated patients |
| Deceleration time < 130 msec | No | Yes | Unassessable in tachycardic patients and in patients with PR interval > 200 msec |
| Lateral E/e′ > 12 | No | Yes | Inaccurate in patients with advanced heart failure |
| B lines on lung ultrasound | No | Yes | Also due to non HF causes |
| NT-proBNP | Yes | No | Elevation also due to non HF causes (caveats), less accurate in obese patients |

## 3. Clinical Pathophysiology of Hypoperfusion

AHF with a clinical presentation of low cardiac output and subsequent organ hypoperfusion is much less common than a congestion profile with normal perfusion [56]. Usually, this condition tends to manifest as overt cardiogenic shock and, therefore, with systolic arterial pressure values < 90 mmHg and mean arterial pressure < 65 mmHg, although in some cases, patients may present with a low output syndrome with more chronic and subacute manifestations related to cellular adaptation to this chronic hypoperfusion state. Once established, hypoperfusion due to low cardiac output (possibly amplified by venous congestion) can adversely affect the function of all organs bringing to a state of multiorgan failure [57]. (Figure 2) The heart can be damaged in AHF due to increased left ventricular pressure and, consequently, parietal stress, increased inotropic and chronotropic sympathetic stimulation [58], and increased afterload due to vasoconstriction, all of which can cause an imbalance between oxygen supply and demand, resulting in myocardial damage (documented by the rise of troponin).

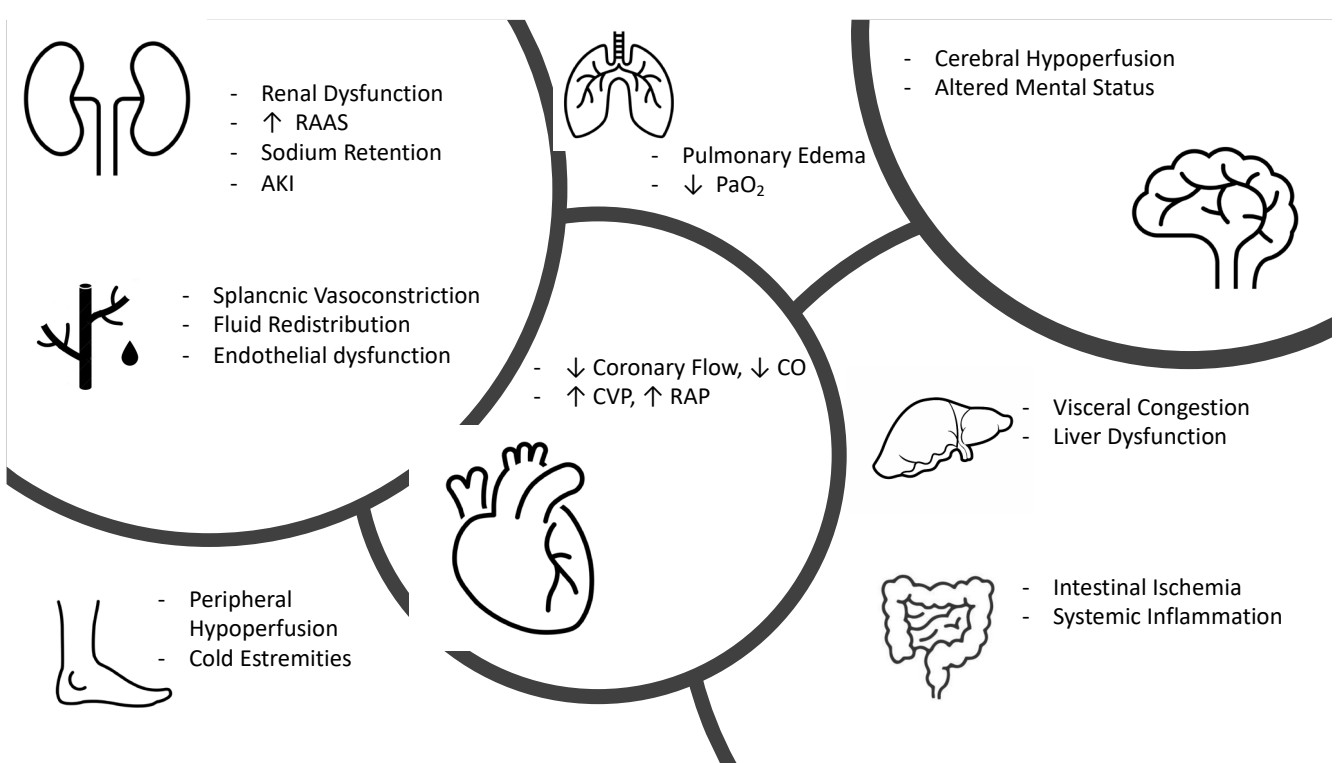

**Figure 2.** Pathophysiology of hypoperfused AHF. RAAS, renin angiotensin aldosterone system; AKI, acute kidney injury; CO, cardiac output; CVP, central venous pressure; RAP, right atrial pressure.

Cerebral hypoperfusion represents one of the earliest manifestations of shock and presents clinically as altered mental status, drowsiness, and dullness [57]. New evidence documents that the intestine is one of the first organs to suffer damage as a result of systemic hypoperfusion with early onset of intestinal barrier ischemia resulting in increased bacterial translocation and release of lipopolysaccharide and endotoxins produced by gram-negative bacteria into the circulatory system resulting in the production of cytokines and increased of inflammation [59,60].

The course of AHF is characterized by normal or even increased vascular volume (in case of peripheral congestion) but with reduced effective arterial blood volume [61].

This initial state of hypoperfusion initially results in acute kidney injury (AKI) that is functional (reversible); however, if the state of hypoperfusion becomes prolonged, it can result in tubular epithelial cell damage with structural (irreversible) renal damage [62].

Hypoxic liver injury (HLI), due to an imbalance between hepatic oxygen supply and demand, can complicate AHF. Generally, this condition is manifested by a marked increase in liver enzymes in the absence of any other known cause of liver injury and, rarely, by severe upper abdominal pain due to liver congestion [63]. Both AKI and HLI represent negative prognostic factors in patients with AHF.

## 4. Pathophysiology-Based Management of AHF

-**Congested and normoperfused patient**: this is the most frequent combination. Supportive therapy is based on intravenous administration of loop diuretics and nitroderivatives. Loop diuretics are the cornerstone of therapy for patients with AHF with pulmonary and/or systemic congestion. In patients with AHF they should be administered intravenously at 1–2.5 times the home dose with an assessment of diuretic response at six hours [30]. In case of inadequate diuretic response (diuresis less than 100–150 mL/h), one can either double the dose of diuretic to be administered intravenously (up to a maximum of 400–600 mg furosemide or 200–300 mg torasemide) or combine metolazone (sequential nephron blockade) [64] to reach a daily diuresis target of 3–5 L [30]. This target should

be maintained until euvolemia is reached. Several reports, indeed, established benefits of discharging patients after a full resolution of congestion [65–67]. However, this is a difficult task to achieve and even more to assess. Many patients are still dismissed from clinics with residual congestion [67,68] with significant higher rates of mortality and rehospitalization [4,69,70]. Clinical evaluation alone, has proven insufficient to examine volume status. Hence, experts recommend using multiparameter-based tools, comprehensive of imaging techniques and NT-proBNP measurements. (Table 2) [30] Even though proBNP-NT is the most studied [71,72] and the only biomarker included in this model, a bunch of novel molecules have been linked with AHF outcomes. Soluble ST2 receptor, expressed when myocardial fibrosis occurs, has been linked with worse prognosis in AHF [73–76]. Growth differentiation factor 15 (GDF15) belongs to TGF-B family has been linked with all cause death and HF hospitalizations (HHF) in secondary analysis of pivotal clinical trials [77–79]. Finally, Fibroblast Growth Factor-23 (FGF23) is a hormone, mostly produced in bones, promoting phosphate excretion managing mineral homeostasis [80]. FGF23 increases during transition of HF from a stable state to a decompensated status and is strictly related with disease severity [81]. However, additional studies are expected to further implement use of these biomarkers in clinical practice.

**Table 2.** Assessment of residual congestion. Adapted from [30]. JVP, jugular venous pulsation, 6MWT: 6-min walk test. BNP: B-type Natriuretic Peptide. NT-proBNP: N-terminal fragment pro B-type Natriuretic Peptide.

| Measurement | Mild | Moderate | Severe |
|---|---|---|---|
| Orthopnea | Absent | Moderate | Severe |
| Hepatomegaly | Absent | Moderate Enlargement | Severe Enlargement |
| JVP | <8 cm | 11–15 cm | >16 cm |
| Edema | Absent | 1 | >+2 |
| 6MWT | >300 m | 200–300 m | <200 m |
| BNP | <100 | 100–299 | >300 |
| NT-proBNP | <400 | 400–1500 | >1500 |
| Chest X-Ray | Clear | Cardiomegaly | - pulmonary congestion—pleural effusion—alveolar edema |
| Vena Cava | None of two:<br>- Max diameter > 22 mm<br>- Collapsibility > 50% | One of two:<br>- Max diameter > 22 mm<br>- Collapsibility > 50% | Both:<br>- Max diameter > 22 mm<br>- Collapsibility > 50% |
| Lung | <15 B Lines | 15–30 B Lines | >30 B Lines |

Vasodilators improve left ventricular performance through venous vasodilatation and thus reduced preload (increased due to congestion) and arterial vasodilatation with reduced afterload [58].

They are used predominantly in the patient with acute vascular type HF who generally has blood pressure values above 140 mmHg [82].

The most widely used are nitroglycerin and nitroprusside, both of which are administered intravenously with low initial doses (10–20 μg/min for nitroglycerin, 0.3 μg/kg/min for nitroprusside) that are subsequently adjusted to the patient's pressor response (up to a maximum dose of 200 μg/min for nitroglycerin and 5 μg/kg/min for nitroprusside) having as target pressors a systolic blood pressure between 90–120 mmHg and a mean blood pressure between 65–70 mmHg [83,84].

Such patients in the absence of high-risk criteria (troponin elevation, worsening renal function) can be managed in intensive brief observation and if responsive to drug therapy do not require hospitalization.

-**Dry and normoperfused patient**: These are generally patients with initial flare-up of chronic HF in whom hospitalization is not indicated but it is sufficient to increase oral therapy. Hospitalization for HF decompensation is often a good time for optimizing guideline-directed medical therapy (GDMT). Most of patients are admitted while on ACE inhibitors/ARBs and beta blockers therapy [56,85,86]. According to recent studies this approach could be safe and effective. In PIONEER-HF patients in sacubitril/valsartan group had reduced rate of HHF and lower levels of NT-proBNP [87]. Additionally, data from TRANSITION supported the feasibility of this approach [88]. Empagliflozin conferred significant net clinical benefit against placebo in EMPULSE study, whether ejection fraction and diabetes [89]. Miller et al. propose a phenotype-based approach, suggesting initiation of low dose mineralocorticoid receptor antagonists in normo-hypertensive patients [90]. Moreover, it is unlike that beta blocker are accountable for decompensation unless they were recently started or uptitrated. Indeed, a recent meta-analysis stated benefit of maintaining beta blocker therapy on death and hospitalizations [91]. Finally, aside prioritizing disease-modifying therapies, we suggest stopping or downtitrating drugs without proven cardiovascular benefit that could impair GDMT tolerance thus facilitating onset of adverse effects such as hypotension.

-**Congested and hypoperfused patient**: These are the most critical patients who need to be managed in the intensive care setting. They can be further divided into two categories according to systolic blood pressure (SBP) [92]:

- SBP > 90 mmHg: the patient benefits from intravenous administration of diuretics and nitroderivatives. It is important to remember that in cases of hypoperfusion, the use of diuretics should be considered after perfusion is restored. If insufficient, the use of positive inotropic drugs such as levosimendan (particularly in patients treated with ß-blockers) or dobutamine should be considered [93].
- SBP < 90 mmHg (cardiogenic shock): we recommend seeing specific readings [60,94].

-**Dry and hypoperfused patient**: hypovolemia should be suspected in these cases, so intravenous fluid administration is useful. A "fluid challenge" [95] can be performed, which is the administration of 250 mL of saline in 15 min and subsequent evaluation of the change in stroke-volume (calculated on echocardiogram) from the baseline value. In patients with an increase in stroke-volume > 10–15%, the reduction in stroke-volume is attributable to the reduction in preload (due to hypovolemia) and consequently adequate hydration therapy should be instituted (in the absence of specific need with an infusion of 25–30 mL/kg/day. saline). (Figure 3).

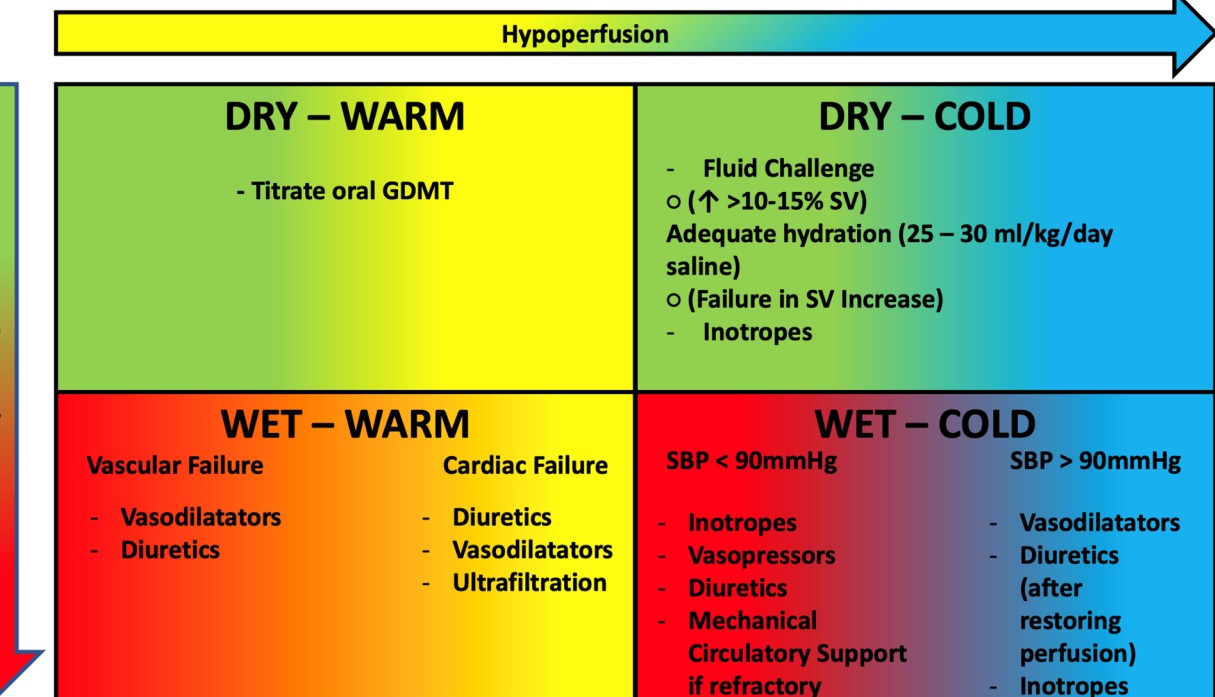

**Figure 3.** Clinical profiles of AHF patients based on congestion and hypoperfusion. See text for further treatment information. GDMT, guideline directed medical therapy; SV, stroke volume; SBP, systolic blood pressure. DBP, diastolic blood pressure. Green: Dry. Red: Congested. Yellow: Warm. Blue: Cold.

In patients with failure to increase stroke volume after fluid challenge (in whom therefore the reduction in output is not preload dependent), the use of inotropes is necessary.

Finally, two aspects often overlooked in common clinical practice should be pointed out:

- Oxygen therapy is not routinely indicated in patients with AHF but only in patients with documentation of hypoxemia ($SPo_2 < 90\%$, $PaO_2 < 60$ mmHg); in such patients, the target to be achieved is a Pa02 between 60 and 90 mmHg [96] (generally corresponding to a $SaO_2 > 90\%$ in chronic hypoxics and a $SaO_2 > 95$ mmHg in other subjects), avoiding hyperoxia that could lead increase peripheral vascular resistance lowering cardiac output [97].
- Disease-modifying drug therapy should be continued in cases of HF flare-ups, except in the patient with hemodynamic instability (symptomatic hypotension or bradycardia, cardiogenic shock), pre-renal acute renal failure, and severe hyperkalemia. In these cases, one should first try to reduce therapy without discontinuing it all together until the patient is stabilized.

**5. Conclusions**

Despite the increasing number of treatment choices for chronic heart failure, people with AHF have not seen the same advancements. AHF is a separate illness with a complex pathophysiology that is still not fully understood and is not being adequately controlled, therefore a large unmet need still weighs on AHF patients. There are significant differences between intravascular and tissue congestion. We suggest that each form of congestion should be treated differently addressing underlying pathophysiology. However, further research is needed to test this hypothesis on hard clinical outcomes.

**Author Contributions:** Conceptualization: D.M., G.P., F.V. and M.V.; writing—original draft preparation: D.M., L.F. and M.L.M.; writing—review and editing: L.F. and D.M. All authors have read and agreed to the published version of the manuscript.

**Funding:** This research received no external funding.

**Institutional Review Board Statement:** Not applicable.

**Informed Consent Statement:** Not applicable.

**Data Availability Statement:** Not applicable.

**Conflicts of Interest:** The authors declare no conflict of interest.

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
