# Peer review of "Pathophysiology-Based Management of Acute Heart Failure"

_clinpract, doi:10.3390/clinpract13010019_

Round 1
Reviewer 1 Report
The authors have summarized the pathophysiology and general authors management of heart failure in 'Pathophysiology-Based Management of Acute Heart Failure: A Narrative Review.'
The article is interesting and connects pathophysiology with the management strategies. Below are a few suggestions to consider:
General:
Authors have used vascular comgestion and pulmonary congestion interchangeably throughput the article. Please consider explaining similarities and/or differences and organize the discussion and conclusions accordingly.
Line 40: Capitalize 'P' in 'Pulmonary congestion'
Figure 1: Replace 'Interstizial' with 'Interstitial' in the figure (at two places within the figure)
Figure 1: 'Lymphatic' is misspelled as 'Linfactic.' Please consider using the correct spellings.
Figure 1: Correct the spellings of 'Interstitial' in the description
Line 95: The authors have mentioned 'recent experimental data' and have not cited any recent study in tbe subsequent references. Reference 35, 36, and 37 are all published prior to 2010. Please consider adding relevant recent studies and add specific description.
Line 95: The authors have mentioned 'human models.' However, the cited studies primarily talk about thoracic aorta from animal models [citation 35].
Lines 95-97:
Authors described that venous congestion plays a role in the pathophysiology of AHF. however, the cite study [35] describes angiotensin II stimulation not venous congestion. Please consider clarifying or edit to include relevant studies.
Line 102-106:
Consider describing the payhophysiology of interstitial edema caused by endothelial cells and dysfunction in the setting of venous congestion.
Line 183-184: Please add reference for diuresis goal
Line 184: Consider describing table as Table. 2
Line 184: Table 2 is mentioned here with no text that is relevant to this table. Please consider adding relevant text or change the location of the text citation of Table 2.
Line 191: The dosage of notroprusside is described as 0, 3 ug/kg/min. 0 ug/kg/min will amount to zero. Please consider adding correct dosages and double-check all other drug dosages mentioned in the paper.
Line 188-209: No references are added to support the suggested management. Pelase consider adding relevant references.
Line 232: Consider changing 'led' to 'lead.'
Author Response
Figure 1: Replace 'Interstizial' with 'Interstitial' in the figure (at two places within the figure)
Figure 1: 'Lymphatic' is misspelled as 'Linfactic.' Please consider using the correct spellings.
Figure 1: Correct the spellings of 'Interstitial' in the description
We have correct indicated spellings
Line 95: The authors have mentioned 'recent experimental data' and have not cited any recent study in tbe subsequent references. Reference 35, 36, and 37 are all published prior to 2010. Please consider adding relevant recent studies and add specific description.
Line 95: The authors have mentioned 'human models.' However, the cited studies primarily talk about thoracic aorta from animal models [citation 35].
Lines 95-97:
Authors described that venous congestion plays a role in the pathophysiology of AHF. however, the cite study [35] describes angiotensin II stimulation not venous congestion. Please consider clarifying or edit to include relevant studies.
Line 102-106:
Consider describing the payhophysiology of interstitial edema caused by endothelial cells and dysfunction in the setting of venous congestion.
[now lines 95 - 115] We have reorganized the statement (we do not talk generically of recent experimental data but first of experimental models, then animal and human models with appropriate citations). Citation 35 now refers correctly only to pro-oxidant stimuli. Moreover we have included new studies (citations 42,43,46,47,48,49 and citation 44-45 published after 2010) and add further description of potential role of endothelium in pathophysiology.
Line 183-184: Please add reference for diuresis goal
Citation 31
Line 184: Table 2 is mentioned here with no text that is relevant to this table. Please consider adding relevant text or change the location of the text citation of Table 2.
We have provided implications behind the importance of using such tool [now lines 193 - 201]
Line 191: The dosage of notroprusside is described as 0, 3 ug/kg/min. 0 ug/kg/min will amount to zero. Please consider adding correct dosages and double-check all other drug dosages mentioned in the paper.
Dosages corrected and checked
Line 188-209: No references are added to support the suggested management. Pelase consider adding relevant references.
We have cited significant papers in support of our suggested management (dry and normoperfused patient citations 86,87,88,89,90,91,92) (congested and hypo perfused patient 93,94,95 ) [now lines 225-250]
we appreciate reviewer 1 suggestions and believe that so we can address areas of the paper that could be improved
Reviewer 2 Report
The review article entitled " Pathophysiology-Based Management of Acute Heart Failure A Narrative Review "review the pathophysiological principles that underlie the clinical symptoms of AHF as well as how to assess them in clinical practice. It is an important topic and the review was well written.
Author Response
The review article entitled " Pathophysiology-Based Management of Acute Heart Failure A Narrative Review "review the pathophysiological principles that underlie the clinical symptoms of AHF as well as how to assess them in clinical practice. It is an important topic and the review was well written.
we really appreciate your opinion on this review article
Reviewer 3 Report
The review "Pathophysiology-Based Management of Acute Heart Failure .A Narrative Review" describes the various pathophysiological aspects of AHF and the possible management of AHF patients.
I congratulate the authors on the manuscript. The text is very clear, as well as the figures. I would suggest to the authors to revise and integrate the paragraph "4. Pathophysiology-based management of AHF". It would be interesting to create a sub-paragraph in which to report in more detail the changes in the levels of biomarkers such as GDF15,sST2, FGF23 as well as the more famous ones mentioned (BNP, NT-proBNP, troponin). Also I would suggest improving the colors of the pattern in figure 3.
Author Response
I congratulate the authors on the manuscript. The text is very clear, as well as the figures. I would suggest to the authors to revise and integrate the paragraph "4. Pathophysiology-based management of AHF". It would be interesting to create a sub-paragraph in which to report in more detail the changes in the levels of biomarkers such as GDF15,sST2, FGF23 as well as the more famous ones mentioned (BNP, NT-proBNP, troponin). Also I would suggest improving the colors of the pattern in figure 3.
As suggested we have included a brief part on biomarkers with relevant citations (72 - 82). Also we have clarified color patterns in figure 3
Reviewer 4 Report
A recent studies with hospitalized patients with heart failure supported the hypothesis that clinical outcomes can be related to patterns of congestion, so some additional information for promotion a personalised pathophysiological-based managment for patients with this syndrome should be crucial.
This paper aims to review the pathophysiology-based management of acute heart failure (AHF) and summarizes the diverse types of a congestion and hypoperfusion.
Pathophysiology-based management of AHF is presented in a concise form and reflects current principles.
Suggestion:
Adding a bendopnea in a list of clinical parameters used for the assessment of congestion in clinical practice (table 1) may improve the utility of the article.
Author Response
Suggestion:
Adding a bendopnea in a list of clinical parameters used for the assessment of congestion in clinical practice (table 1) may improve the utility of the article.
As suggested we have included bendopnea in tab.1